# Factors Shaping the Lived Experience of Resettlement for Former Refugees in Regional Australia

**DOI:** 10.3390/ijerph17020501

**Published:** 2020-01-13

**Authors:** Laura Smith, Ha Hoang, Tamara Reynish, Kim McLeod, Chona Hannah, Stuart Auckland, Shameran Slewa-Younan, Jonathan Mond

**Affiliations:** 1Centre for Rural Health, University of Tasmania, Launceston, TAS 7250, Australia; thi.hoang@utas.edu.au (H.H.); tamara.reynish@utas.edu.au (T.R.); stuart.auckland@utas.edu.au (S.A.); jon.mond@utas.edu.au (J.M.); 2School of Social Sciences, University of Tasmania, Launceston, TAS 7250, Australia; kim.mcleod@utas.edu.au; 3School of Health Sciences, University of Tasmania, Launceston, TAS 7250, Australia; chona.hannah@utas.edu.au; 4Translational Health Research Institute and School of Medicine, Western Sydney University, Campbelltown, NSW 2560, Australia; S.Younan@westernsydney.edu.au; 5Centre for Mental Health, Melbourne School of Population and Global Health, University of Melbourne, Carlton, VIC 3053, Australia

**Keywords:** refugees, resettlement, lived experience, social environment, refugee health, public health, quality of life, health services, housing, qualitative research, regional and rural Australia

## Abstract

Refugees experience traumatic life events with impacts amplified in regional and rural areas due to barriers accessing services. This study examined the factors influencing the lived experience of resettlement for former refugees in regional Launceston, Australia, including environmental, social, and health-related factors. Qualitative interviews and focus groups were conducted with adult and youth community members from Burma, Bhutan, Sierra Leone, Afghanistan, Iran, and Sudan, and essential service providers (*n* = 31). Thematic analysis revealed four factors as primarily influencing resettlement: English language proficiency; employment, education and housing environments and opportunities; health status and service access; and broader social factors and experiences. Participants suggested strategies to overcome barriers associated with these factors and improve overall quality of life throughout resettlement. These included flexible English language program delivery and employment support, including industry-specific language courses; the provision of interpreters; community events fostering cultural sharing, inclusivity and promoting well-being; and routine inclusion of nondiscriminatory, culturally sensitive, trauma-informed practices throughout a former refugee’s environment, including within education, employment, housing and service settings.

## 1. Introduction

The United Nations High Commissioner for Refugees (UNHCR) [1] (para. 1) defines a refugee as “a person forced to flee their country because of violence or persecution.” People fear persecution because of their nationality, religion, race, social group membership or political opinion, with the leading causes of refugees fleeing countries including ethnic, religious and tribal violence, and war [2]. By the end of 2017, displacement due to armed conflict, violence or persecution occurred for 68.5 million people worldwide. Of these, 25.4 million were classified as refugees. With continuing global conflict, numbers are expected to rise, as circumstances driving displacement persist [2,3]. Resettlement refers to the transference of refugees from a state where they have sought protection to a different state in which they have been granted permanent residence status [4]. To our knowledge, only two studies have specifically examined resettlement experiences of refugees in Australia [5,6] and only one of these [5] was conducted in a regional area. This is concerning given resettlement in regional and rural areas is likely to be particularly challenging, with service access restricted by distance and a maldistribution of the health workforce [7]. The high number of refugees resettling in Australia (e.g., 180,790 entrants between 2009 and 2018 [8]), highlights the importance of providing adequate resettlement support. To do this, accurate and timely information regarding resettlement experiences is needed to develop and deliver appropriate services.

In terms of studies which have specifically looked at the experience of resettlement, Sweeney [5] explored the resettlement experiences of Sierra Leonean former refugees in regional Launceston, Australia. Issues found to be associated with people feeling at home included experiences with settlement services, marginalization, racism and family conflict, gender issues, employment issues, financial status and connection with local communities. Social inclusion was dependent on whether experiences were acknowledged and recognised. In addition, Vromans et al. [6] found that for Afghan and African refugee women living in south-east Queensland, a major city area [9], social connection, financial independence, emotional wellbeing and self-efficacy were essential to effective resettlement, citing a need to further explore these experiences and the strategies needed to improve social networks.

Other studies have explored additional factors affecting the lives of refugees in Australia, touching on those affecting resettlement. Atwell, Gifford and McDonald-Wilmsen [10] found that for refugees from Sudan, Burma, Afghanistan, Bosnia and Liberia in the city of Melbourne, Australia, having to learn a new language affected resettlement. Colic-Peisker and Walker [11] found that for refugees from Bosnia, Serbia, Croatia, Ethiopia, Oromo, Sudan, Eritrea and Somali resettled in Perth, Australia, the struggle to shed the “refugee identity” affected resettlement, and the authors acknowledged the importance of understanding the effects of past trauma on current day experiences, however not exploring these effects further. Colic-Peisker [12] found that employment discrimination impacted overall life satisfaction for refugees in Perth, Australia, and Correa-Velez, Gifford and Barnett [13] found that for refugee youth in Melbourne, Australia, the factors that were most likely to affect resettlement included those which promoted a sense of belonging and hominess, and provided opportunities to become part of society. As described by Marmot et al. [14], having access to secure housing is an important social determinant of health. In a systematic review of relevant studies by Ziersch and Due [15], findings suggested that in countries of resettlement, there were often issues with insecure housing arrangements and tenure, affordability and suitability.

Impacts of the migration experience include increased risks to health and adverse effects on psycho-social functioning associated with adjusting to a new environment [3,13,16]. Studies mostly focused on former refugees’ pre-arrival circumstances [17,18,19,20], with less thought to the experience of resettlement, with its overwhelming obstacles [21] and current realities. Several areas are therefore highlighted as in need of further investigation, including a need to explore and acknowledge experiences of resettlement [5], to pick up where previous studies have fallen short by exploring how trauma impacts experiences of resettlement [11], as well as looking at how discrimination may be experienced in employment and in accessing housing [12,15], particularly in regional areas where barriers to access to health and support services are pronounced [22].

The concept of lived experience is understanding human experiences, and how these are influenced by subjective factors including race, class, gender and religion [23]. The current study aimed to explore the lived experience of resettlement for former refugees in Launceston, a regional area of Australia. It aims to offset the scarcity of existing qualitative health and social research in this area, drawing on perspectives of former refugees as well as the essential service providers (ESPs) who work with former refugees through their everyday provision of services.

## 2. Materials and Methods

### 2.1. Setting

The study was conducted in Launceston, a regional city of around 84,000 people, located in the north-east of the island state of Tasmania, Australia [24]. Between 2006 and 2016, around 3124 former refugees resettled in Launceston, primarily from Bhutan, Nepal, Afghanistan, Sudan, Eritrea, Liberia, Burma and the Congo [25].

### 2.2. Research Design

A qualitative research design was used to explore, in-depth, refugee experiences of resettlement, including those related to culture and place [26,27]. Further, a descriptive phenomenological approach was utilised in this study [28,29] to focus on the commonality of lived experience within a group, with the goal of arriving at an explanation of the nature of the phenomenon in question [30]. Consistent with this theoretical framework, focus group discussions allowed insight into how participants constructed shared meaning on the topic of resettlement [31]. In order to create a supportive environment for participants to share their experiences, focus groups were composed of members from the same communities and conducted in a location that was familiar to them (the Migrant Resource Centre, Launceston, Northern Tasmania (MRC)).

### 2.3. Participants

A criterion sampling [32] approach to participant recruitment was employed, with inclusion criteria being either a former refugee who arrived under the Australian Refugee and Humanitarian Program or an ESP who provided everyday services to former refugees in Launceston, being at least 14 years of age (to allow for youth representation), and current residency in Launceston.

### 2.4. Procedure

Former refugee participants were recruited through the MRC, the primary provider of resettlement support services to refugees in Launceston. People were invited to participate through their membership in pre-existing cultural community groups including the Afghan Women’s and Men’s Friendship Groups, and the MRC’s Youth Advisory Group. Participants from Burma, Bhutan, Sierra Leone, Afghanistan, Iran and Sudan were targeted, these being the main refugee groups living in Launceston according to MRC data [25]. ESPs included government services, health service providers, for example employees from the local public hospital and a counsellor at a local counselling service for migrant communities, and English language program providers.

Six focus groups and seven individual interviews were conducted among a total of 31 participants representing a range of ethnic and cultural backgrounds and service providers (Table 1). Youth Advisory Network former refugee participants (*n* = 6) were aged between 14 and 18 years old, half were male, and half were female. All ESP participants and adult former refugee participants were aged 18 years or older, with just over one-half (52%) of these being female, and the remainder male. To protect the anonymity of participants, further breakdown of sex and age is unable to be disclosed.

The research team worked with the MRC to refine research aims and study methodology and to assist with participant recruitment, scheduling interviews and focus groups.

Two strategies were employed to maximise ecological validity of findings [33]. First, data triangulation saw data from multiple sources, including in-depth individual interviews and focus groups with ESPs and with former refugees, used to clarify emergent themes and explore the phenomenon of resettlement [34]. Second, data interpretation was facilitated by means of peer debriefing, whereby two members of the research team reviewed the data and verified links between data and derived themes.

Data saturation informed the number of interviews and focus groups required, where data was continuously analysed throughout collection, and additional collection and/or analysis was deemed unnecessary where novel themes ceased to appear [35]. In addition, sessions were conducted by two research team members, one leading discussions and managing sessions, the other providing support. Sessions were in English or, where interpretation was required, Bicultural Support Workers (BSWs) were utilized, as was the case for five out of the 13 focus groups/interviews. Sessions with former refugee participants took place at the MRC and with ESPs at either their workplace, the University of Tasmania, or the MRC. Sessions, which utilised a schedule of semi-structured questions (Table 2), lasted on average 70 min and were audio-recorded.

### 2.5. Data Analysis

Where utilised, translation by BSWs was as close as possible to the original account, incorporating nuances of language and discussion context. Sessions were transcribed verbatim by the research team, with data integrity confirmed by a second researcher through listening to recordings and checking against written transcriptions. Participant names were replaced with identification numbers to ensure confidentiality. Consent forms were signed by participants prior to sessions, provided in English. Where required, a BSW provided interpretation, relaying questions to the research team for resolution prior to consent being obtained. Ethics approval was granted by the Tasmanian Social Sciences Human Research Ethics Network (no. H0017137), and data were collected from April to July 2018.

Data were analysed using the six phases of thematic analysis [36] and were read and re-read to search for meaning and patterns. Notes taken assisted coding, and NVivo qualitative data analysis software (QSR International Pty Ltd., Doncaster, Australia, Version 10, 2014) was used to store, code, classify and sort data. Data were initially coded and then sorted into potential themes. Themes were reviewed and refined to identify relationships, defined, then named. Data were then interpreted. Two researchers analysed data independently, and results were compared and discussed at regular meetings until consensus was reached. Reporting was guided by the consolidated criteria for reporting qualitative research (COREQ) [37].

## 3. Results

Four main themes emerged from the data, revealing life aspects considered most crucial to resettlement experiences: English language proficiency; employment, education and housing environments and opportunities; health status and health service access; and the quality of broader social life.

### 3.1. English Language Proficiency

Participants emphasised the importance of English to their resettlement experiences. The relationship between the two extended across daily life, including accessing employment, education and services:


*“I believe that language is the most important thing that we need to get, to be able to settle.”*
*—former refugee participant—Afghan men’s group*


*“It’s hard to find a job…the language barrier that…stop us getting that work.”*
*—former refugee participant—adult community leader*


*“…not enough recognition of what barriers these kids are hitting with English…they get put into a class and they basically have to swim…”*
*—ESP participant*

ESP participants noted the health implications which flowed from not having English language proficiency:


*“…if you can’t speak your issues then you won’t access the service….”*
*—ESP participant*


*“We had an incident, not, not long ago, where one of my youth students, had an accident and he hurt himself… “what did you do, did you call an ambulance?”, “no”, “why not?”, “ …we didn’t have the language to be able to call an ambulance” ….”*
*—ESP participant*

Participants were cognisant of the challenges former refugees face in learning English. Mental health issues and trauma impacted the ability to learn English. Participants who left family behind in their countries of origin experienced worry, guilt and feeling out of control. One former refugee participant referred to this constellation of feelings as “forward trauma”:


*“…where people who settle here have an incredible guilt about their family members back home, they’ve left behind…brings incredible feelings for people here, responsibility and guilt.”*
*—ESP participant*

ESP participants indicated how trauma informed the learning capacities of former refugees:


*“[the] impact of what we would call trauma on their cognitive abilities and the ability to concentrate and focus and, and actually learn …”*
*—ESP participant*

Age was another potential barrier to learning, with adults finding it harder to learn than youth, more pronounced in adults who lacked previous learning experience:


*“Since they had no education when they were in Iran/Afghanistan, so it was difficult for them to start English here.”*
*—former refugee participant—Afghan women’s friendship group (spoken through BSW)*


*“…adults really seem to struggle. Certainly the ones, the former refugees that were literate in their own language seemed to grasp it better…”*
*—ESP participant*

Participants emphasised the significance of becoming proficient in English to positive resettlement experiences. However, this intersected with desires to retain the language and cultural traditions associated with the former refugees’ countries of origin. There were concerns that practicing speaking English at home would supplant peoples’ parent language and impact younger generations:


*“…talking in English, if we forget Nepali, so our children, our brothers, sisters, they will won’t be honouring our tradition, culture…So, we need to preserve our language, culture, tradition…”*
*—former refugee participant—adult community leader*


*“…if your children don’t speak Farsi, if your children don’t have knowledge of your culture, I don’t think that’s going to go down very well when they go back for holidays…”*
*—ESP participant*

Inappropriateness and inflexibility of study materials and program delivery were cited as barriers to learning English. Learners were not provided with study materials in formats they could access readily outside of class:


*“…students get, ‘here is a printed paper’ every day…we never go back and read what we have studied before so it’s a daily basis teaching...”*
*—ESP participant*


*“…with [name of charity organisation]…when you finish the first 800 h…you have to have a break of 6 months before you can attend another 800 h. So he said…‘when I came back to get another 800 h I have to start from the start’.”*
*—ESP participant*

Recent government changes to program providers were cited as another barrier to English language acquisition:


*“…the Australian government…tendered out to the cheapest tender… language classes which [education provider] were running originally and then [employment services provider] won the tender for, they were not prepared…people say, “I’ve been here seven months and I have not had an English class yet, I want to be a member of this community, how can I do that without speaking English?”*
*—ESP participant*

### 3.2. Employment, Education, and Housing Environments and Opportunities

Accessing employment, education and suitable housing was considered essential to successful resettlement for both adult and youth former refugees. For some communities, employment was noted as being fruitful and the types of work in these areas similar to those back in home countries:


*“…they are all employed. People are working even those who have never been to school…those who are not working, there’re probably only 2 people that I know they are not working out of the whole Sudanese community so that’s quite a big success…”*
*—former refugee participant—adult community leader*


*“…things like fruit picking… seasonal work…that they enjoy doing, that’s familiar to them, because a lot of them come from rural areas as well, originally.”*
*—ESP participant*

Overall, accessing employment proved challenging in regional Launceston due to limited employment opportunities and not being able to speak English. In addition, participants spoke of additional barriers including discrimination and the nontransferability of qualifications:


*“…they’re conscientious workers, they’ve probably worked their backside off than most Australians, but because of that prejudice they don’t get a chance.”*
*—ESP participant*


*“…they changed their name to western names and they call them for interview so when they went, the people were surprised you know. So that was just to prove that there are forms of discrimination…”*
*—former refugee participant—adult community leader*


*“I studied architecture, I believe it’s a universal standard studying architecture…but I found it difficult….”*
*—former refugee participant—Afghan men’s group*

Participants offered several suggestions for tackling these barriers, including incentives to employ refugees, industry-specific English language programs or “functional English” and businesses employing former refugees in front-of-house positions.


*“…state and the federal organizations have to break the ice. They should be the ones that should start employing Africans. When you go to Melbourne, you go to Centrelink, you got a lot of Africans working there. … put more emphasis on giving the incentives to these community organizations and businesses to be able to employ people from refugee migrants….”*
*—former refugee participant—adult community leader*


*“…do something about functional English…if I came here, I was involved in a program that was working with people from the regional areas… focused in learning English using…everything that has to do with farming… learning was connected directly with their occupation.”*
*—former refugee participant—adult community leader*

Participants recognised the benefits of education for themselves and their community. In contrast to the challenges of accessing employment, participants were able to positively compare education opportunities in Launceston with those of the past:


*“I am very happy living in Australia, because the women, refugee in Iran and others can’t study in a school because the government in Iran they say “you are a refugee, you can’t study in here…””*
*—former refugee participant—Youth Advisory Network*


*“…if I can study here then I can become good person and also, I can help to other my community for example if I’m doctor in future I would help my community or Australian people, I will be very happy to help them…”*
*—former refugee participant—Youth Advisory Network*

Adults regarded learning as a positive experience, however, they also faced difficulties with accessing education where the inflexibility of programs led to clashes with familial responsibilities:


*“My life is really different here so when I was in Iran there wasn’t any school, there wasn’t any opportunity to learn… I’m going to school and learning, so feeling really happy.”*
*—former refugee participant—Afghan women’s friendship group*


*“…she’s been very busy with raising her children and… she has to start it all [English learning] over again, after these years.”*
*—former refugee participant—Afghan women’s friendship group (spoken through BSW)*

Refugee students faced further discrimination by being assigned to levels of education based on age, rather than ability, which placed them on an uneven playing field from the onset of their education:


*“…kids who are going to college and high school they get treated the same way. For example he said his cousin been assessed based on her age, they said ‘you are 15 so you should go grade 10 or 11′ for example, and he said all of a sudden she found herself in an education system that she had no idea before, she’d never studied for example geography before, and she had to compete with other student who been in the system for all their life…”*
*—former refugee participant—Afghan men’s group (spoken through BSW)*

To support refugee students, ESP participants also highlighted the importance of teachers being trauma-informed and culturally aware, to ensure this playing field was more equitable:


*“… if you’re understanding where the behaviours are coming from and you know what makes them feel safe… [you] create a welcoming environment in a classroom…”*
*—ESP participant*


*“…if you’ve got a classroom of 30, you might have 5 refugee kids that you probably got 15 to 20 kids in there who have experience some kind of trauma so we’re not just talking about you know the refugee kids’ population…”*
*—ESP participant*


*“A lot of the schools are now are including cultural…education to the rest of the class, including things like ‘hey, it’s Afghan new year let’s do this in the classroom for the Afghan kids and everyone join in and learn about that’… make people feel more comfortable and welcome…”*
*—ESP participant*

ESPs spoke of the dream that former refugees had of owning their own home in Australia, and of saving and managing budgets and sharing accommodation to make this happen:


*“… [they] just want to be homeowners. I think that’s the huge goal, for a lot of them, they want to own their own their little piece of Australia, and they want to own a home”*
*—ESP participant*


*“…might be two or three families that live together in a household so they’re pooling their money because they’re a collectivist community, that’s what they’re used to doing, so that’s how they can save money…”*
*—ESP participant*

Many participants cited the discriminatory practices that were inherent with accessing rental accommodation, from assumptions being made based on their family size and ethnicity, to discriminatory practices imbedded within policies which did not accommodate not being able to understand English:


*“…many people came with their families, big families from 4, 5 to 8, will struggle to get a house. You see, they apply for houses and people for some reasons they thought they will not, they probably destroy the houses or there are many other things that house owners think of…sometimes they are stuck, they cannot, because they find it difficult to find a house themselves because of a lot of things that the owners of the house and if you are competing with a local person here…”*
*—former refugee participant—adult community leader*


*“…they can’t read and write, you know, they don’t know what is the information they got from real estate so they live for, you know, until 12 months, they haven’t find, you know, new house or, so they have to move, you know, within maybe 2 ah 14 days and they really struggling…”*
*—former refugee participant—adult community leader*

Housing accessibility was also discussed in terms of the suitability of geographical area and the rising costs associated with needing to feel safe and live among other members of familiar communities, as one ESP explained:


*“…it’s really really important that they feel they’re in the safe… they’re all buying up Newnham [Launceston suburb], unfortunately they’ve pushed the prices sky high on each other, but anyway, because that’s where they feel safe, it’s a safety, a feeling of safety there because their community is there…”*
*—ESP participant*

The effects of not being able to access employment were also reflected in living arrangements, where as a result of not being employed and having suitable financial means, people were often forced to live with other people, particularly family, which was not necessarily ideal for social cohesion of families and the wellbeing for those individuals:


*“…example of a young man… arrived here when he was about 21…back in his country he had been working in a construction job since he was about 15 which enabled him to earn some money and have his own life and his own social all of that kind of stuff and it also enabled him to escape some of the violence and not very nice things that were happening in his home…his family all got given their refugee status together as a family and sent here and were given a house and so now he is back in living with his family…They don’t particularly like each other. They don’t have…financial means to move out separately. They don’t have you know he doesn’t have friends here who he could move out with or spend all of the time with to avoid being at home...”*
*—ESP participant*

### 3.3. Health Status and Access to Health Services

Health status and access to health services were identified as crucial dimensions of resettlement. A supportive, welcoming service environment affected resettlement experiences, with simple gestures often making the biggest impact:


*“…we always used to in the clinic, go out of our way to learn the words for hello, goodbye, really simple…And they used to love that… simple things like that, make them feel really welcome.”*
*—ESP participant*

ESP participants shared their impressions of how previous experiences impacted present health and functioning, including how trauma impacted ability to learn English, parenting and children’s mental health. This was particularly emphasised where the process of interpreting was seen as traumatic for some children:


*“…many of the former refugees with mental health illnesses, probably all tied up with, you know, experiences they had before they came to Australia….”*
*—ESP participant*


*“…there are still a lot of intergenerational trauma stuff that comes from where parents not being in a safe enough space… So not being able to form a secure attachment because they were constantly worried about what might happen or where the next meal is going to come from…”*
*—ESP participant*


*“We have to use one of our accredited interpreters, and that’s both to protect the children, because I know that can be very traumatic…”*
*—ESP participant*

ESP participants noted the persistent and pre-existing chronic health conditions among older people arriving in Australia, due to being unable to access health services:


*“…with the more recent arrivals in the last few years, we’ve seen these issues like long-term issues come through. Um, things that people require ongoing, chronic health conditions that really affect people’s lives and most of them need a carer as well, so I think that that’s a huge issue.”*
*—ESP participant*


*“…from people who have experienced many, many, many years of refugee camp…Bhutanese, Burmese communities, that the older people in their communities, their health is really, really poor, there’s lots of diabetes, respiratory issues, somatic body issues, arthritis, yeah, so quite debilitating health issues…”*
*—ESP participant*

Health service access was identified as an important dimension of resettlement with experiences often compared to health service access in previous home countries:


*“…in the longer term…health will be much better because they have access to all these services that are free, whereas in Nepal where the Bhutanese refugees come from… services just weren’t available...”*
*—ESP participant*


*“…in my country, you know, growing up in the countryside there is no doctor, there’s no nurse so we have to walk, four hours, six hours. So, in comparison…here is really, really good …”*
*—former refugee participant—adult community leader*

Participants identified lacking culturally informed practices across health services, with assumptions made based on cultural background, and the impact of the normalisation of westernised medical models of treatment:


*“…there’s a lack of understanding of the different ways that mental health might be treated or might be addressed or might be understood other than just to depression or anxiety.”*
*—ESP participant*


*“…when they were in the camp they went to a doctor and said, “I’m sad” and the doctor said “cool, here take these tablets, um it will fix it” and they get here…and they say, “I don’t want counselling, I just want tablets.””*
*—ESP participant*

Lack of English posed a significant barrier to service access. Lack of interpreters provided and frequent misunderstanding was referred to as “systemic racism”, affecting the likelihood of former refugees engaging with health services:


*“…people are treated differently, and they are seen as differently and, in this system, here, in this hospital system… I see so many human rights violations, for instance, not getting an interpreter…so I think there’s a continuum of racism, and I think there is systemic racism…”*
*—ESP participant*


*“…because they cannot speak English with just brush them off when they say something, you know the little things that people face…they don’t really want to go there again…”*
*—former refugee participant—adult community leader*

Participants referred to health services using children as interpreters or “language brokers” for their parents. This practice was supposed to be banned by organisational policy, and placed children in compromised positions and decreased the likelihood that important health information was being relayed effectively. Other systemic barriers to access included difficulties with appointment bookings and wait times and insufficient resources in heath service departments to support former refugees.

### 3.4. Quality of Broader Social Life

As mentioned, many participants stated that being connected to community and having people from their home countries living nearby was essential to feeling supported throughout resettlement:


*“…community is the backbone so that when there is a, someone is in trouble, like they should be cooperative, understand the problem…those who have arrived before more than 4, 5 years like here, we are from the same community and also our language is the same…”*
*—former refugee participant—adult community leader*

In addition, community leaders were crucial for aiding and coordinating support:


*“…quite often it is the community leaders who will be the first point of call for pretty much anything.”*
*—ESP participant*

Support from the Launceston community also helped women, youth and men alike, with feelings of connection and safety. Feelings of inclusion and belonging were strengthened from being included in local community and social activities:


*“When I arrived here, when my family arrived here, we had 10 people that came to the airport just to say welcome. So, it impressed us a lot, because there was a lot of disrespect and discrimination in Iran… happy with the people’s behaviours and manners here...”*
*—former refugee participant—Afghan women’s friendship group*


*“…they’ll take a group of women who are into sewing, or spinning or weaving…and they’ll invite them along to share…And you know end up working together, or cooking.”*
*—ESP participant*

The MRC, government services, religious and community organisations and volunteers were all cited as helping participants with resettlement and support:


*“…I cannot think of any other organisation in this city provided help the way MRC does...”*
*—ESP participant*


*“…we shouldn’t be forgetting the help of the Centrelink that we’ve received and the settlement journey.”*
*—former refugee participant—Afghan men’s group*


*“…the church was also something that is very important to me that helped me to settle…”*
*—former refugee participant—adult community leader*

Participants also discussed the challenges associated with creating a social life in a place with differing cultural and social norms. Participants experienced frustrations with Launceston’s lack of neighbourliness and isolated living arrangements in the community:


*“…we had neighbours, but we hardly see them or even if they are outside…they hardly talk to us…for us it was quite difficult because we lived in smaller communities and everybody knew everybody…”*
*—former refugee participant—adult community leader*

Frustrations were also noted with the western toilet system, and with accessing familiar foods. Many participants compared their current experiences with positive experiences of their home countries, leading to sadness and worsening of overall mental health and resettlement. Differences in gender and family roles were noted, with women and young people taking more responsibility within their families:


*“…especially the Afghan community have…very specific gender roles… it’s very instilled. Over time though things tend to relax a bit more.”*
*—ESP participant*


*“…men found it very difficult to lose the position in the family where they make the decisions…”*
*—former refugee participant—adult community leader*

Some participants feared cultural traditions might become lost as younger generations and children experienced pressures to perform in education and employment in the Australian context, as well as carry on cultural traditions:


*“…young people are adapting to Australian culture differently and parents are really worried that they will lose things that their former culture and young people having to do that balance over “what is really important to me in my home culture what kind of things do I wanna keep”...”*
*—ESP participant*


*“…young people feel a bit like a utility, so they are the ones that have to realize all of their parents’ expectations for coming…they have to really well at school and become doctors… it’s kind of their job to take the opportunities that Australia offers but also to carry on their cultural traditions...”—ESP participant*
*—ESP participant*

Former refugees felt that assumptions were often made based on how they looked, their nationality or religion:


*“Because they don’t know the story why we migrated here. That’s the main problem.”*
*—former refugee participant—adult community leader*


*“…in every country, in every place, there are few people who may think differently from other people. Some people think we’ve come to take their jobs…because we have jobs and they don’t…”*
*—former refugee participant—adult community leader*


*“…differences in cultural practices and religious practices between his religion and culture and Australia’s existing religion and culture…”*
*—former refugee participant—Afghan men’s group (spoken through BSW)*

In addition, ESPs also noticed this discrimination and racism when interacting with former refugees, explaining this was often exacerbated by stereotypical viewpoints, the media and fear mongering:


*“…a tendency in humans to mistrust things that are different and people that are different…prejudice largely fueled by the media of course…especially place like Tasmania which is very, Anglo, very insular…”*
*—ESP participant*


*“…people have seen on the media that a single Muslim man has, you know, blown up this…you get this diatribe about you know, they shouldn’t be allowed to come to Tasmania, they’re all terrorists and, you know…”*
*—ESP participant*


*“…fear mongering against Muslims for instance, and this is what I get scared about here, about people coming here…I just get a bit scared for them, because there have been racist attacks here, violent, racist attacks…”*
*—ESP participant*

Some participants experienced bullying at their housing establishment, however, when comparing experiences to those in previous countries, they were still thankful for being here:


*“…some neighbours they start bullying from their place. Where I live…there are 2 boys, 19 and 20 years…and they start bullying or saying something, swearing…”*
*—former refugee participant—adult community leader*


*“…comparing the experience of a migrant first in Iran and a migrant person in Australia, was incomparable. That the level of discrimination they faced in Iran luckily wasn’t here in Australian society…”*
*—former refugee participant—Afghan men’s group (spoken through BSW)*


*“Even though the bogan at the supermarket might give them a filthy look or you know tell them to go back to where they came from…that’s great compared to where they’ve come from…”*
*—ESP participant*

## 4. Discussion

This study explored the lived resettlement experiences of a diverse group of former refugees in regional Australia. Thematic analysis of these experiences revealed four areas that are crucial to resettlement, namely English language proficiency; employment, education and housing environments and opportunities; health status and health service access; and the quality of broader social life.

Findings concerning the effects of English language proficiency converged with those reported by Atwell et al. [10] in their study of former refugees living in the city of Melbourne, where English was a common factor discussed by all participants, influencing every aspect of resettlement, including accessing education, employment and health services. However, new insights from the current research further elucidate how this systemic discrimination is experienced by many former refugees and the role it plays in hindering essential aspects of resettlement, with services failing to consistently offer language interpretation and to provide appropriate resources to obviate this disadvantage. Participants offered suggestions to help overcome discriminatory practices associated with learning English, including providing language programs that are flexible to individual needs, which consider lifestyle and familial responsibilities, along with experiences of trauma and individual factors like age and learning experience. Additionally, it was suggested that resources should be provided in accessible formats (workbooks, online content or videos, rather than print outs) to ensure content complements individual learning styles. Participants noted that where a change of service provider disrupts people’s ability to learn, funders of these services, that is, government, should ensure that tender processes are equitable, and that service quality is prioritised over cost. Further exploration of English language programs is warranted to determine what formats, for example online or face-to-face delivery, are effective for refugee populations in regional and rural areas.

The importance attached to employment opportunities by participants is consistent with findings by Sweeney [5], who found meaningful employment is essential for effective resettlement in regional areas, as in metropolitan areas. In addition, the current research highlights additional barriers to employment related to lack of English language proficiency, discrimination, prejudice and the nontransferability of qualifications. Overcoming employment barriers starts with government and other services leading by example and normalising the employment of refugees. This is particularly relevant to regional areas like Launceston, where the population is predominantly from English ancestry [24] and minority population groups, including, but not limited to, former refugees, face additional challenges accessing employment opportunities. The provision of English language courses tailored to specific industries in demand in regional areas would likely be beneficial in this regard. Career mentoring for recently arrived former refugees could focus on matching existing skills sets and qualifications with job roles and career pathways, thus helping to alleviate the effects of system-level barriers and qualification nontransferability [38]. Government funding schemes could provide local employers with incentives for employing refugees, such as through subsidised apprenticeships, particularly for in-demand trades which are failing to attract local job seekers. Initiatives of this kind need to include measures designed to reduce prejudice and incentivise employment among local employers.

Educational opportunities and quality in regional Launceston were highly valued by participants and seen as a means to ensure that individuals are set up for a good life, as has also been reported in previous research [39]. The greatest perceived barriers to accessing education among participants in the current study were the education system structure and learners being assigned to grade levels based on their age, rather than ability, including their English language proficiency. This was seen to set the learner up for failure from the outset. Additionally, no account was seen to be taken of the impact of prior trauma, social dislocation and related factors on learning capacity [40,41,42]. As suggested by Thomas [43], the human right to education dictates the use of inclusive, culturally sensitive teaching practices capable of accommodating experiences of trauma and cultural and individual differences in the expression of this. A “one size fits all” approach to education for former refugees is neither appropriate nor equitable [42,44]. Additional support for resettled students prior to starting formal education should be mandatory, aiding with English learning, and learning the processes involved with formal learning environments, thus enabling the learner to be set up for success rather than failure.

Participants spoke of the dream of owning their own homes, as a product of resettling, while relying on temporary rental accommodation in the meantime. While accommodation support services do exist in Australia, including regional areas like Tasmania [45,46], access to suitable housing may be impeded by discriminatory practices and the English language proficiency needed to understand rental documents. As also described by Ziersch [15], this including assumptions being made due to a person’s family size or ethnicity, applicants not having sufficient English to be able to understand essential rental and contractual documents, and preference being given to local applicants. Given almost one in ten refugees are faced with primary homelessness and are forced to live with friends or family to avoid this occurrence [47], the importance of addressing English language proficiency, employment, and enforcing legislation and equal rights/nondiscriminatory practices are essential. This could include making local government responsible for coordinating volunteers to help refugees navigate local housing options upon resettlement, as suggested by Rose and Charette [48]. Other incentives could focus on private landlords and/or include schemes to ensure equitable access for refugees, utilising interpreters or retrofitting existing housing to accommodate larger families, for example.

The impact of English language proficiency and discriminatory practices on access to health services was also highlighted by participants in the current study. Risks involved with using children to interpret medical information in appointments ranged from misinformation to negative mental health affects experienced by the child. While similar findings have been reported in previous studies of Chinese immigrant children in Canada and Latino immigrant children in the United States [49,50], the current findings highlight their salience in a regional setting. Participants in the current study also noted that the use of child interpreters continued despite the existence of organisational policies precluding this practice. This suggests a failure of policy enforcement and an area in need of further exploration. 

The current findings also build on previous studies, including those with Iraqi, Afghanistan and Iranian refugee men as well as Somali, Russian and Kurdish immigrants [51,52,53], highlighting the interplay between issues of trauma, distress and help-seeking behaviour—and issues relating to “mental health literacy” more generally—as these are experienced by former refugees. In addition, this relates to health professionals’ sensitivity and competency in managing these and related issues, including treatment adherence and self-monitoring [54,55]. Potential solutions to these challenges, mentioned by participants in the current study and in previous research, include ensuring adequate cultural competency and trauma-informed training for employees across health services and employing health professionals from Culturally and Linguistically Diverse backgrounds [55], or providing mandatory refugee-specific, trauma-informed training for all employees across health services [56]. Such strategies can help ensure former refugees are afforded the time required to adapt to cultural differences and to process experiences of trauma, while receiving care in their parent language. Gaining cultural competence will enable health professionals to understand how former refugees may interpret health, especially mental health, and the additional “extra-therapeutic” factors that have been identified as influencing their well-being [57]. Furthermore, investigation into current levels of cultural competence and trauma-informed practices within rural and regional health services would be welcome.

The last factor to emerge, namely, the quality of participants’ broader social life, was seen to play a major role in how welcome they felt on arrival and throughout the first few years of resettlement. Participation in community events, both within cultural groups and within the local community more broadly, was highlighted as a significant factor in effective resettlement, providing opportunities to share cultures, promote social inclusion and create networks [11,12,13]. The degree to which a person’s culture and social norms aligned with those of the wider community was seen to dictate how comfortable a person felt living there. This was associated with gender and family roles, living arrangements and access to familiar foods. Discrimination was seen to affect not only access to education, employment and health services, but also various other aspects of daily life, including access to housing. At the same time, participants reflected on the fact that their experience of discrimination seemed relatively mild when compared with what they had experienced prior to resettlement. There is a paucity of research informing which sorts of programs and strategies are most likely to be effective in improving the quality of resettled refugees’ social lives.

Limitations of the current research include the use of BSWs for interpretation for just over one-third (38.5%) of all the focus groups/interviews, which may have limited access to understated nuances in meanings and wording. In addition, the use of convenience sampling methods may have detracted from the representativeness of the sample and, in turn, the generalizability of the findings. Strengths of the research include participants recruited representing a range of cultural backgrounds, and interviews and focus groups delivered face-to-face, ensuring the establishment of rapport and effective communication.

## 5. Conclusions

This article builds on existing studies by exploring the resettlement experiences of former refugees from multiple perspectives. It provides new insights to inform policies relating to health, housing and resettlement services for refugees resettled in regional areas. This includes the need for accessible English programs, for trauma-informed, culturally sensitive service provision and for better enforcement of policies designed to preclude the use of children as interpreters. We hope that the findings will provide an incentive to further research in this field, particularly quantitative research, to inform the development of interventions and improve resettlement experiences.

## Figures and Tables

**Table 1 ijerph-17-00501-t001:** Focus group and interview participant numbers and characteristics.

	No. Participants	Group Membership	Background	Bicultural Support Worker Utilised? (Yes/No)
Group 1	5	Women’s Friendship	Afghan	Yes
Group 2	4	Women’s Friendship	Afghan	Yes
Group 3	3	Leader’s Group	Bhutanese	Yes
Group 4	4	Men’s Group	Afghan	Yes
Group 5	6	Youth Advisory Network	Afghan, Iranian	Yes
Group 6	2	Essential Service Provider (ESP)	Launceston Library	No
Interview 1	1	ESP	Centrelink	No
Interview 2	1	ESP	Launceston General Hospital (LGH)	No
Interview 3	1	ESP	LGH	No
Interview 4	1	Leader’s Group	Sudanese	No
Interview 5	1	ESP	Phoenix Centre (Counselling service)	No
Interview 6	1	Leader’s Group	Burmese	No
Interview 7	1	Leader’s Group	Sierra Leonean	No

**Table 2 ijerph-17-00501-t002:** Focus group and interview topic guide.

**Questions for Former Refugees**
1. How have you found resettling in Launceston?
2. Do you feel a sense of belonging in Launceston? Does Launceston feel like home to you?
3. How has life been for you since you’ve resettled? How has it been for family and friends here?
4. What has helped you the most settling into your life in Launceston? What has helped you the least?
**Questions for ESPs**
1. What do you think are the overall experiences for former refugees who have resettled in Launceston?
2. What do you think are the three biggest resettlement benefits encountered by former refugees in Launceston? What are the three biggest barriers to resettlement?
3. From your work with former refugees in Launceston, what sort of supports have you found former refugees use? What sort of things make former refugees feel welcome/not so welcome, upon arrival?
4. How has learning a new language been experienced by the former refugees you work with? What experiences have you noticed for family members who are required to act as interpreters?

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
