# Peer review of "Factors Shaping the Lived Experience of Resettlement for Former Refugees in Regional Australia"

_ijerph, 2020, doi:10.3390/ijerph17020501_

Round 1

Reviewer 1 Report

The research topic of this paper (Manuscript ID: ijerph-680825) is interesting. The extant literature indicates that research on the factors influencing the lived experience of resettlement for former refugees remains relatively scarce. To address this problem, this paper uses a qualitative design to explore the above problem. The above method proposed by this paper is to be encouraged. Despite all this, there are several possible revisions as follows: (1)Why only choose 31 subjects to participate into this qualitative step? How to make sure the theoretical saturation? (2) Academic contribution is essential for each paper. Please provide the relevant content.

Author Response

Why only choose 31 subjects to participate into this qualitative step? How to make sure the theoretical saturation? 

As noted in the revised manuscript (p.4), data saturation informed the number of interviews and focus groups required, where data was continuously analysed throughout collection, and additional data collection and/or analysis was deemed unnecessary where novel themes ceased to continue appearing. A reference is provided for the use of this approach.

Academic contribution is essential for each paper. Please provide the relevant content. 

Author contributions are provided on p.13 of the revised manuscript.

Reviewer 2 Report

The present manuscript reports the lived experience of resettlement for former refugees in regional Launceston, Australia, and suggest factors that primarily influence the resettlement. The topic is is important, study is well written and should be of interest to readership of IJERPH. I have some concerns mentioned below:

The essential service providers (ESPs) provided information from their own experience based on working with refugees. Is that comparable with the information provided directly from refugees? Was there any differences? It should not be mentioned in study limitations? How many interviews have been done by BSWs? I think it’s also important and should be mentioned in the study limitations. Line 583 - “Strengths of the research include a relatively large sample size with participants recruited to represent a range of cultural backgrounds”. 31 subject is relatively large compared to studies on resettlement experiences of refugees in Australia but I don’t think that it is a large sample size - although all subjects are refuges, they have different cultural backgrounds, as you mentioned.

The refugees health and integration into society is a very important topic and I would suggest to conduct quantitative research that would be probably more powerful for policy makers.

Author Response

The essential service providers (ESPs) provided information from their own experience based on working with refugees. Is that comparable with the information provided directly from refugees? Was there any differences? Any differences should be mentioned in study limitations? 

We agree that this warrants consideration. Where accounts were comparable these have been highlighted in the manuscript to support the theme that was identified. For example, in the Results section of the revised manuscript (p.6), barriers to accessing employment was presented from both the perspectives of the ESPs and the former refugees. However, and as is clear from Table 2 (revised manuscript, p.4), ESPs and former refugees were asked different questions, such that differences in responses would be expected. Hence these differences are more something to note and comment on where appropriate than a limitation of the study.

How many interviews have been done by BSWs? I think it’s also important and should be mentioned in the study limitations. 

Agreed and amended accordingly – revised manuscript p.3 (Table 1), p.13.

Line 583 - “Strengths of the research include a relatively large sample size with participants recruited to represent a range of cultural backgrounds”. 31 subject is relatively large compared to studies on resettlement experiences of refugees in Australia but I don’t think that it is a large sample size - although all subjects are refuges, they have different cultural backgrounds, as you mentioned. 

We agree that use of the term “relatively large sample size” could be seen to be misleading and we are happy to delete it in the text of the revised manuscript (p.13).

The refugees health and integration into society is a very important topic and I would suggest to conduct quantitative research that would be probably more powerful for policy makers

We agree that it will be important to use the current findings to inform quantitative research and we are happy to note this in the Conclusion of the revised manuscript (p.13).